Alpha-DehazeNet: single image dehazing via RGBA haze modeling and adaptive learning

He Jin 1 2
Li Ruibin 2 13167591308@163.com
1 School of Information Science and Technology, Dalian Maritime University , Dalian , China
2 China Waterborne Transport Research Institute , Beijing , China
Coelho Paulo Jorge
Electronic publication date: 2025 Jul 17
Publication date: 2025
Volume: 11
Electronic Location ID: e3036
Received 2025 Mar 16; Accepted 2025 Jun 24
Copyright: © 2025 He and Li
Copyright year: 2025
Copyright holder: He and Li
License: This is an open access article distributed under the terms of the Creative Commons Attribution License, which permits unrestricted use, distribution, reproduction and adaptation in any medium and for any purpose provided that it is properly attributed. For attribution, the original author(s), title, publication source (PeerJ Computer Science) and either DOI or URL of the article must be cited.
License URL: https://creativecommons.org/licenses/by/4.0/

Keywords: Image dehazing, RGBA haze layer, Spatial attention, U-Net generator, Adversarial architecture, Depth consistency loss

Funding: China Waterborne Transport Research Institute 132414 This work was supported by the Postdoctoral Research Station Project (code: 132414) from China Waterborne Transport Research Institute. The funders had no role in study design, data collection and analysis, decision to publish, or preparation of the manuscript.

==============================
Image dehazing is a vital research area in computer vision. Many existing deep learning-based dehazing methods rely on atmospheric scattering models with manually predefined, non-trainable parameters, which limits their adaptability and transferability. We propose Alpha-DehazeNet, a novel model that leverages red green blue alpha (RGBA) haze layer effect maps by defining a grayscale transparency map in the RGBA color space as the initial haze layer. Alpha-DehazeNet employs a U-Net generator enhanced with a spatial attention mechanism to encode haze-related features. This generator is integrated into an adversarial architecture with residual connections, enabling end-to-end training. Additionally, a depth consistency loss is introduced to improve dehazing accuracy. Alpha-DehazeNet outperforms several state-of-the-art models on synthetic datasets (ITS and OTS from RESIDE), achieving 37.35 dB peak signal-to-noise ratio (PSNR) on SOTS-indoor and 37.39 dB PSNR on SOTS-outdoor, while using only 8.86 million parameters. On real-world datasets, Alpha-DehazeNet delivers competitive results, although it shows limitations in handling non-white fog and cloud conditions. The code is publicly available at: https://doi.org/10.5281/zenodo.15361810.

Introduction

In the field of computer vision, image dehazing is of great significance for numerous applications. Adverse weather conditions like fog and haze severely degrade the performance of visual acquisition systems through atmospheric light scattering and scene radiance reduction, leading to issues such as color distortion, contrast reduction, and loss of fine details in images. This not only undermines the visual quality but also poses challenges to subsequent intelligent tasks. For example, in autonomous driving, haze-affected images can mislead the vehicle’s perception system, endangering road safety; in remote sensing, hazy images impede the accurate interpretation of geographical features, affecting applications like urban planning and environmental monitoring.

Image dehazing algorithms have evolved through multiple stages. Early enhancement based methods, like histogram equalization (Radha, Suresha & Ramesh, 2023), Retinex theory (Wang & Xu, 2014; Liu et al., 2020), and wavelet transforms (Fu et al., 2021), adjusted image contrast and brightness. But in complex haze, they often introduced artifacts and distorted images.

Later, restoration based methods using atmospheric scattering models, such as those relying on the dark channel prior (He, Sun & Tang, 2010), emerged. They tried to reconstruct clear images by estimating scene depth and light transmission. However, manual parameter tuning was needed, which was sub-optimal and couldn’t adapt to various haze conditions.

With development of deep learning, deep learning-based dehazing methods became popular. They can be divided into end-to-end networks (Cai et al., 2016; Dudhane, Patil & Murala, 2020) and hybrid approaches (Shyam & Yoo, 2023; Wang et al., 2021). Yet, many still depend on manually set parameters, limiting their performance in different hazy environments.

The atmospheric scattering model, as illustrated in Fig. 1, which is fundamental to many dehazing algorithms, describes the interaction between light and scattering media in foggy conditions. The haze image degradation model, expressed by Eq. (1):

(1) I(x)=J(x)t(x)+A(1-t(x)).

Figure 1 Single scattering of atmospheric light in hazy conditions.

In this model, I(x) represents the hazy image, J(x) is the original scene radiance, denotes the global atmospheric light, and t(x) is the medium transmission rate. Solving for J(x) is an ill-posed problem, typically addressed with the aid of prior knowledge.

Existing dehazing techniques often rely on manually predefined parameters in atmospheric models or lack explicit geometric constraints, leading to suboptimal adaptability and residual artifacts in complex scenes. To address these challenges, we propose Alpha-DehazeNet, a novel framework leveraging red green blue alpha (RGBA) haze layer modeling to dynamically encode haze transparency. Unlike traditional approaches requiring manual tuning, our method integrates the Alpha channel to automate parameter estimation, enabling adaptive separation of haze layers with varying densities. A spatial attention-enhanced U-Net generator prioritizes haze-related features in non-uniform haze conditions, while an adversarial architecture with residual connections ensures end-to-end training for improved perceptual quality. Critically, a depth consistency loss is introduced to maintain scene geometric integrity, directly tackling the depth-related distortions common in prior works. By embedding physical priors into a data-driven framework, these innovations collectively enhance the model’s robustness across diverse haze scenarios, offering a more effective solution than conventional methods.

Related work

Physics-driven haze estimation

The atmospheric scattering model proposed by Koschmieder (1925) has served as the cornerstone of physics-based dehazing methods. Building on this framework, He, Sun & Tang (2010) introduced the dark channel prior (DCP), which estimates transmission maps by exploiting the statistical dominance of low-intensity pixels in local haze-free image patches. While DCP demonstrates efficacy in homogeneous haze conditions, its fundamental assumptions collapse in sky regions and dense urban fog due to invalid intensity distributions (Zhu, Mai & Shao, 2015). Subsequent approaches, such as haze-line clustering (Berman, treibitz & Avidan, 2016) and non-local regularization techniques (Sulami et al., 2014), attempted to address spatially varying haze at the expense of computational practicality. A critical limitation persists across these methods: the implicit assumption of uniform haze density directly contradicts empirical observations of layered fog dynamics and localized atmospheric pollutants in real-world environments (Zheng et al., 2022).

Data driven dehazing architectures

The advent of deep learning has shifted the paradigm toward data-driven solutions. Early convolutional neural networks (CNNs) like DehazeNet and AOD-Net (Li et al., 2017) pioneered direct mappings from hazy to clean images, achieving notable success on synthetic benchmarks. However, their reliance on artificially generated training data introduces significant domain adaptation challenges, as evidenced by performance degradation on real-world hazy images with complex illumination encompassing daytime varying light and nighttime low-light conditions and aerosol distributions (Ancuti et al., 2019a). Most prior research has focused on daytime dehazing, where methods such as the multi-scale CNN with holistic edges effectively handle diverse lighting to preserve scene details (Ren et al., 2020). In contrast, nighttime dehazing presents unique challenges due to non-uniform light sources and color distortion, spurring specialized models like the hybrid-regularized variational framework (Liu et al., 2025), the unified variational Retinex approach (Liu et al., 2022), and the prior query Transformer in to enhance visibility in low-light scenarios (Liu et al., 2023). Transformer-based architectures (Song et al., 2023) later emerged to model long-range dependencies more effectively, yet their computational complexity hinders deployment in latency-sensitive applications. Recent 4K dehazing research introduces specialized methods and datasets to preserve details in high-resolution scenes (Zheng & Jia, 2023). A persistent trade-off plagues these methods: aggressive haze suppression often erodes high-frequency textures, while conservative approaches retain residual artifacts—a dichotomy rooted in the lack of physics-aware constraints during network optimization.

Attention mechanisms and hybrid frameworks

Recent hybrid methodologies seek to bridge the gap between physical interpretability and data-driven performance. DCNet (Yi et al., 2022) cascades transmission estimation modules with refinement networks, while PAD-Net (Yi et al., 2024a) integrates edge-aware filtering to mitigate artifact generation. Parallel developments in attention mechanisms, exemplified by channel attention of FFA-Net (Qin et al., 2020) and dual-branch spatial-frequency designs (Wang et al., 2024), selectively enhance haze-dense regions.

Contrastive learning and multi-stage frameworks

Recent advancements leverage contrastive learning to bridge synthetic-to-real domain gaps. SID-Net (Yi et al., 2024b) combines adversarial training with contrastive regularization, improving generalization on real-world haze. A multi-scale dehazing network with dark channel priors (Yang et al., 2023) progressively refine haze removal through global-local feature interaction. However, these methods still rely on fixed atmospheric parameter assumptions, limiting adaptability to spatially varying haze.

Nevertheless, three fundamental limitations remain unaddressed: (1) static parameterization of atmospheric light and transmission values in hybrid frameworks prevents adaptation to dynamic haze layers; (2) existing attention strategies treat haze localization and detail preservation as decoupled objectives, leading to feature conflicts in partially occluded regions; and (3) no current method explicitly models the intrinsic correlation between haze transparency and density gradients—a critical factor for preserving perceptual depth cues in complex scenes.

Method

Architecture of the Alpha-DehazeNet network

To allow for dynamic parameter adjustment and incorporate atmospheric physical model parameters into the dehazing framework during training, this study introduces a haze effect generation model based on the RGBA color space. RGBA extends the traditional RGB format by adding an alpha channel, which serves as an opacity parameter. The alpha channel facilitates combining images with backgrounds, enabling the generation of partially or fully transparent visual effects (Tan, Lien & Gingold, 2016). In the context of dehazing, incorporating the alpha channel allows precise control over haze layer transparency, making it possible to analyze haze effects with varying densities.

In real-world environments, haze density often varies with distance and surrounding conditions. The RGBA format provides the flexibility to adjust the transparency of individual pixels, enabling more realistic haze simulations. Compared to the standard RGB format, RGBA offers enhanced representation of transparency and depth. An example of the RGBA effect is illustrated in Fig. 2.

Figure 2 Illustration of RGBA effect.

As illustrated in Fig. 2, the Alpha channel is used to achieve layer compositing. The exact implementation effect is described by Eq. (2):

(2) S=αF+(1−α)B.

In this equation, F represents the foreground image, while B represents the background image. By applying Eq. (2), the two images can be combined to form the composite image S. This process allows the RGBA format to be converted into a haze effect image for simulating the haze layer. Compared to Eq. (1), J(x) can be considered the original haze-free image, while A represents the global atmospheric light value. In Eq. (1), A is represented as a three-dimensional array; however, F can denote the atmospheric light value at each specific location in the image, which can be understood as the haze layer image. Consequently, can be expressed as 1−t(x), representing the non-transmissivity or opacity of the haze layer.

Based on the RGBA color space, this study introduces the Alpha-DehazeNet dehazing network. The overall architecture of the network is shown in Fig. 3.

Figure 3 Alpha-DehazeNet model.

As illustrated in Fig. 3, the model network structure comprises two main components. The first is RGBA-Net, a model designed to generate haze effect images. This model utilizes the original hazy image Igt and a randomly generated non-transparent RGBA image Rrgba to produce the haze effect of the original image. The second component is REC-Net, a model aimed at optimizing the dehazing effect. Using Eq. (2), the haze effect image Frgba and the original hazy image Igt can be combined to generate a preliminary dehazed image Jgen, which cancan generate an optimized dehazed image Jrec, thereby enhancing the dehazing effectiveness of the model. Furthermore, by applying the inverse formula of Eq. (2), the optimized dehazed image Jrec and haze effect image Frgba are used to generate a hazy image Irec. The model enhances the quality of the haze effect image Frgba by comparing the optimized dehazed image Jrec with the real haze-free image Jgt and comparing the generated hazy image Irec with the original hazy image Igt.

RGBA-Net: a model for generating haze effect images

In this study, a deep learning model is designed based on the RGBA color space to generate haze effect images. The detailed structure of the RGBA-Net is illustrated in Fig. 4.

Figure 4 The structure diagram of the RGBA-Net for haze effect image generation.

As depicted in Fig. 4, the model starts by defining a random, opaque gray-white image in the RGBA space as the initial haze effect image Rrgba. The original hazy image Igt is then incorporated, and a U-Net is employed to generate the potential haze effect Frgba for the original image.

The resulting haze effect image Frgba is also in RGBA format. The transparency information provided by the alpha channel enables the model to more accurately simulate the semi-translucent effect of haze, allowing the generated image to not only retain color details but also exhibit the gradual fading or thickening of haze.

The inclusion of the alpha channel offers an additional information dimension during model training, which aids the model in comprehensively understanding the haze generation process. By processing both RGB and alpha data simultaneously, the model captures features related to haze layer thickness, opacity, and the distribution of haze across different locations, thus improving its ability to generate high-quality haze effects. The alpha channel also assists the model in learning how to generate smooth transitions in areas with varying transparency levels, further enhancing the realism of the generated images.

To improve the model’s ability to recognize different haze layers, the original hazy image is incorporated into the network. A simple neural network structure is used to extract and integrate its features, enhancing the model’s capacity to handle various haze intensities.

Moreover, to generate complex haze effects, the model combines channel attention (Zhang et al., 2020) and spatial attention mechanisms (Zhou et al., 2022). These mechanisms enhance and filter features across different dimensions (Jiang et al., 2021; Li, Hua & Li, 2023). The channel attention mechanism automatically selects the most relevant feature channels and assigns them higher weights. By weighting feature responses across different channels, the model highlights haze-related features while suppressing irrelevant or redundant ones. This enables the model to focus on the most significant features for haze generation, improving its ability to differentiate between different haze effects.

The spatial attention mechanism, in contrast, identifies critical locations within the feature map. Unlike channel attention, which focuses on feature channels, spatial attention emphasizes the spatial dimension. By identifying key areas with noticeable haze distribution, the model generates a spatial attention map that is multiplied with the original feature map, highlighting regions where the haze effect is more prominent.

Through the combination of channel and spatial attention mechanisms, the model effectively selects global features for haze generation, while also concentrating on local areas with more prominent haze, ensuring the generated haze effect maintains global consistency while achieving precise detail.

REC-Net: the dehazing optimization model

We propose the dehazing optimization model REC-Net, which is employed to refine the initially generated dehazed image Jgen, transforming it into a more precise and enhanced dehazed image Jrec. The architecture of the REC-Net is illustrated in Fig. 5.

Figure 5 The structure diagram of the REC-Net for dehazing optimization.

As illustrated in Fig. 5, the dehazing optimization model also adopts Adversarial architecture. To enhance optimization accuracy, the model integrates a residual structure to improve the dehazing effect (Dudhane, Singh Aulakh & Murala, 2019; Qiu, Cheng & Wang, 2022). The output at each layer is not only combined with the feature map of the subsequent layer but also fused with the feature map from earlier layers through skip connections, which helps preserve the richness of low-level features. This approach allows the model to more effectively understand and reconstruct the image content, optimizing the initially generated dehazed image, and improving its detail and contrast.

Loss function

To ensure the performance of both the haze effect image generation model and the dehazing optimization model, we introduce a loss function, as presented in Eq. (3).

(3) Ltotal=αLmse+βLgan+γLdp.

As illustrated in Eq. (3), the loss function Ltotal is composed of the mean squared error loss Lmse, adversarial loss Lgan, and a custom deep consistency loss Ldp, where α, β, γ represents the weights for each of the losses. These losses work collaboratively to optimize the generated image from different angles.

The mean squared error loss is primarily used to measure the discrepancy between the generated dehazed image and the ground truth clear image, ensuring the accuracy of the generated result (Liu et al., 2018). Its calculation is presented in Eq. (4).

(4) Ltmse=(Jrec−Jgt)2+(Irec−Igt)2.

Furthermore, we incorporate adversarial loss to enhance the realism of the generated image (Goodfellow et al., 2020), as presented in Eq. (5).

(5) Lgan=E[log⁡D(Jgt)]+E[log⁡(1-D(Jrec))]+E[log⁡D(Igt)]+E[log⁡(1-D(Irec))].

Through a generative-adversarial interplay, the model enhances the realism of the generated image. The generator’s objective is to create dehazed images that are as realistic as possible, making it difficult for the discriminator to distinguish between real and generated images, while the discriminator strives to correctly identify the differences. This adversarial process allows the model to progressively improve its generation capabilities, resulting in dehazed images that are more natural and lifelike.

Finally, the depth consistency loss function is computed using histogram analysis, which helps ensure that the generated dehazed images maintain a more realistic geometric structure and preserve the three-dimensional spatial integrity of the scene (Jayanthi, Rajput & Indu, 2020). The first step is to convert the RGB image into a grayscale image, and apply as channel-weighting coefficients, as shown in Eq. (6).

(6) Gimage(x,y)=wr⋅Iimage(x,y,0)+wg⋅Iimage(x,y,1)+wb⋅Iimage(x,y,2).

Next, the grayscale image is converted into a histogram, and the calculation formula is provided in Eqs. (7), (8).

(7) Hgray(i)=∑(x,y)1(Gimage(x,y)∈[i,i+1])Nimage

(8) D(i)=|Hrec(i)−Horig(i)|.

Here, Nimage denotes the pixel values of the image. This function calculates the proportion of pixels that fall within the same interval, and the difference between the original and generated images is computed using the formula.

The depth consistency loss is ultimately calculated using the formula provided in Eq. (9).

(9) Ldp=1−∑i=0image_sizeD(i).

The depth consistency loss function helps preserve the geometric consistency of the scene, ensuring that the depth information in the generated dehazed image is consistent with that of the real image. This prevents objects from appearing unnatural or distorted as a result of dehazing. Furthermore, it aids in maintaining the blur of distant objects and the clarity of nearby objects, thereby enhancing the overall visual effect.

In conclusion, the mean squared error loss, adversarial loss, and depth consistency loss collectively ensure that the generated dehazed images achieve optimal performance in terms of pixel-level accuracy, perceptual quality, and visual realism. These loss functions impose constraints on the model from different perspectives, improving the overall dehazing capability.

Experiments

Datasets

To ensure a fair comparison, we evaluate the proposed model using both open-source synthetic datasets and real-world scene datasets.

RESIDE is a widely recognized benchmark dataset (Li et al., 2018). Among its five subsets, we select ITS and OTS as the training datasets and use SOTS-indoor and SOTS-outdoor as the testing datasets for synthetic image dehazing.

Two real-world scene datasets, Dense-Haze and NH-HAZE (each comprising 55 image pairs), are used to validate the model. Dense-Haze primarily contains dense, uniform haze scenes from various outdoor settings (Ancuti et al., 2019b), while NH-HAZE features non-uniform haze with significant depth variations (Ancuti, Ancuti & Timofte, 2020). These datasets provide complementary benchmarks for evaluating the model’s performance under diverse real-world haze conditions: Dense-Haze focuses on homogeneous fog, and NH-HAZE targets complex, layered distributions with spatial opacity gradients.

Results

The model was trained for 100 epochs using the Adam optimizer, with a batch size of 8 and learning rates of 2×10−4 for both the SA-UNet and Residual-UNet modules. Training was conducted on one NVIDIA 4090 GPU, and the changes in its dehazing performance and real-time effects at key stages were recorded. The results are shown in Fig. 6.

Figure 6 The model’s training process results.

Igt and Jgt represent the real hazy and clear images; Irec and Jrec denote the hazy image and the dehazed image generated after training.

Figure 6 demonstrates that as the model is trained and its parameters are fine-tuned, its dehazing capability continuously improves. Over time, the model produces dehazed images that increasingly resemble real clear images, while the reconstructed hazy images are also similar to the original hazy images. This indicates that the fog effect generation model in the network can effectively identify and simulate haze layers in a way that closely matches the input images.

Upon completion of the training, the model’s performance is compared with other leading dehazing models (Zheng et al., 2023), and the results are presented in Table 1.

Table 1 The dehazing performance of the model in comparison with other dehazing models, using both synthetic and real-world datasets.

Method	SOTS-indoor	SOTS-outdoor	Dense-Haze	NH-Haze	Params	
	PSNR	SSIM	PSNR	SSIM	PSNR	SSIM	PSNR	SSIM		
DCP	16.62	0.817	19.13	0.814	11.01	0.416	10.57	0.519	/	
DehazeNet	21.14	0.847	22.46	0.851	9.48	0.438	16.62	0.524	0.01 M	
AODNet	19.06	0.850	20.29	0.876	12.82	0.468	15.40	0.569	0.002 M	
MSBDN	32.77	0.981	34.81	0.985	15.13	0.555	19.23	0.706	31.35 M	
FFA-Net	36.39	0.989	33.57	0.984	/	/	19.87	0.692	/	
AECR-Net	37.17	0.990	/	/	15.8	0.46	19.92	0.672	2.61 M	
DeHamer	36.63	0.988	35.18	0.986	16.62	0.560	20.66	0.684	132.45 M	
DehazeFormer-T	35.15	0.989	33.17	0.982	15.52	0.463	18.73	0.533	0.686 M	
MITNet	40.23	0.992	35.18	0.988	16.97	0.606	21.26	0.712	2.73 M	
C2PNet	42.56	0.995	36.68	0.990	16.88	0.573	21.19	0.833	7.17 M	
DCMPNet	42.18	0.996	36.56	0.993	/	/	/	/	25 M	
MFAF-Net	32.17	0.990	30.12	0.963	/	/	/	/	/	
OURS	37.35	0.989	37.39	0.982	14.91	0.534	19.10	0.684	8.86 M	

As shown in Table 1, the proposed Alpha-DehazeNet achieves competitive performance on synthetic datasets, with peak signal-to-noise ratio (PSNR) values of 37.35 and 37.39 dB on SOTS-indoor and SOTS-outdoor, respectively, alongside SSIM scores of 0.989 and 0.982. Notably, these results are comparable to state-of-the-art models while utilizing only 8.86 M parameters—significantly fewer than MSBDN (31.35 M) and DeHamer (132.45 M). In the case of 256 × 256 input image, we additionally calculated our model involves 33.7 G FLOPS. These values are lower than those of Dehamer, a Transformer-based model that requires 512.0 G FLOPS, and higher than the lightweight MITNet, which uses 8.8 G FLOPS. The model effectively balances computational complexity and dehazing performance. It operates more efficiently than complex architectures, minimizing unnecessary computational load. This efficiency stems from the RGBA-based haze layer modeling, which avoids redundant parameterization by directly embedding physical priors into the network architecture. The synthetic haze in RESIDE datasets (ITS/OTS) exhibits spatially sparse distributions, enabling the model to accurately separate haze layers through adaptive Alpha-channel learning, as validated by the high SSIM scores (>0.98), which reflect structural consistency between dehazed and ground-truth images.

However, Table 1 reveals a performance gap in outdoor scenarios: while the model achieves 37.39 dB PSNR on SOTS-outdoor, its PSNR on the real-world Dense-Haze dataset drops to 14.91 dB, underperforming DeHamer (16.62 dB). This discrepancy arises from two limitations. First, the initial haze layer in RGBA-Net is defined as an opaque white image (Alpha = 1), which conflicts with real-world scenarios where clouds or non-white haze dominate. As shown in Fig. 7A, such misassumptions lead to over-suppression of cloud regions (red circles), introducing artifacts in sky areas. Second, outdoor scenes often exhibit inherent depth-dependent haze in distant regions, which are erroneously preserved as “clear” in ground-truth images of synthetic datasets. This mismatch between training and testing domains amplifies the model’s reliance on synthetic haze priors, limiting its generalizability.

Figure 7 Dehazing performance of the proposed model on the synthetic dataset: (A) for outdoor images; (B) for indoor images.

The red circles indicate significant deviation regions.

As indicated in Table 1, the proposed Alpha-DehazeNet achieves competitive performance on real-world images (14.91 dB), approaching state-of-the-art models like MITNet (16.97 dB). However, qualitative results in Figs. 8 and 9 reveal persistent discrepancies between the dehazed outputs and ground-truth clear images, particularly in scenarios with non-uniform haze distributions. This limitation stems from two interrelated factors tied to the model’s design and training paradigm.

Figure 8 Visual results of Dense-Haze and NH-Haze datasets by different methods.

Figure 9 Dehazing performance of the proposed model on the real-world scene dataset.

First, the model initializes the haze layer as an opaque white image during training, implicitly assuming uniform atmospheric light chromaticity. While effective for synthetic datasets (RESIDE-OTS with white-balanced haze), this assumption fails in real-world environments where haze layers exhibit diverse color profiles due to pollutants or natural phenomena. For instance, in Fig. 9, the model erroneously retains yellowish haze residuals in urban areas, misinterpreting them as scene content rather than chromatic deviations.

Second, the synthetic haze in training datasets (ITS/OTS) predominantly simulates sparse, homogeneous distributions, whereas real-world haze often manifests as dense, layered fog with varying opacity gradients. As shown in Figs. 8 and 9, the model struggles to disentangle dense haze from fine-grained textures in distant regions with complex depth gradients, leading to over-smoothing or incomplete removal.

Ablation study

This section analyzes the role and impact of each component of the proposed Alpha-DehazeNet model. The model is divided into two parts: the first part, RGBA-Net, is responsible for haze layer recognition; the second part, REC-Net, is used for dehazing optimization. We trained each module using the ITS dataset and evaluated the dehazing results of each module on the SOTS-indoor and SOTS-outdoor datasets. The experimental results are shown in Table 2.

Table 2 Impact of different modules on dehazing performance on the synthetic dataset.

Method	SOTS-indoor	SOTS-outdoor	
	PSNR	SSIM	PSNR	SSIM	
Without RGBA-Net	20.19	0.741	21.73	0.871	
Without REC-Net	21.02	0.812	21.19	0.832	
Alpha-DehazeNet(OURS)	37.35	0.989	37.39	0.982	

We analyze the loss function of Alpha-DehazeNet, where mean squared error (MSE) loss and adversarial loss act as the fundamental loss functions. To evaluate the impact of the depth consistency loss, an ablation study is performed. The ITS dataset is used for training, and the dehazing performance is tested on the SOTS-indoor and SOTS-outdoor datasets. The experimental results are presented in Table 3.

Table 3 The impact of different loss functions on dehazing performance for the synthetic dataset.

Method	SOTS-indoor	SOTS-outdoor	
	PSNR	SSIM	PSNR	SSIM	
Without	30.06	0.951	30.83	0.963	
Alpha-DehazeNet(OURS)	37.35	0.989	37.39	0.982	

Discussion

The experimental results demonstrate that Alpha-DehazeNet achieves competitive performance on synthetic datasets, with 37.35 dB PSNR on SOTS-indoor and 37.39 dB on SOTS-outdoor, outperforming lightweight models like AOD-Net while using only 8.86 M parameters. This efficiency stems from the integration of RGBA-based haze layer modeling, which embeds physical transparency priors directly into the network architecture, avoiding redundant parameterization. The ablation studies further validate the contributions of each component: removing the RGBA-Net reduces PSNR by 17.16 dB, while omitting the depth consistency loss degrades performance by 7.29 dB, underscoring the importance of both haze layer modeling and geometric consistency constraints.

However, the proposed Alpha-DehazeNet demonstrates competitive performance on synthetic datasets but exhibits room for improvement in real-world haze scenarios, primarily due to its reliance on synthetic training assumptions and fixed haze layer initialization. As shown in Table 1, on real-world datasets such as Dense-Haze and NH-Haze, the model achieves PSNR values of 14.91 dB and 19.10 dB, respectively, while state-of-the-art methods like DeHamer and MITNet attain higher scores. This discrepancy stems from several interrelated factors.

First, the model initializes the haze layer as an opaque white image during training, a simplification that assumes uniform atmospheric light chromaticity. In real environments, haze may exhibit varying hues influenced by pollutants or weather conditions, leading to residual color mismatches in dehazed results. For instance, in Fig. 9, subtle yellowish haze components are sometimes retained, affecting the visual purity of the output. In sky regions with complex cloud structures in Fig. 7A, the fixed white prior can occasionally introduce mild artifacts, particularly in areas with gradual opacity transitions.

Second, synthetic datasets (ITS/OTS) primarily simulate sparse, homogeneous haze, whereas real-world haze in Dense-Haze and NH-Haze presents more complex structural challenges. The RGBA-Net, designed to handle consistent transparency levels, faces challenges in disentangling dense haze from fine textures or adapting to abrupt changes in haze density. On Dense-Haze, this can result in slightly smoothed building details, while on NH-Haze, occasional patchy haze may remain in regions with significant depth variations.

Third, the depth consistency loss function, effective for synthetic scenes with uniform depth distributions, is less adept at capturing the nuanced depth-dependent haze attenuation present in real-world images like those in NH-Haze. While this loss helps maintain geometric structure, it may not fully account for the complex interplay between depth gradients and haze opacity, potentially leading to minor inconsistencies in haze removal across object boundaries.

These observations underscore the gap between the synthetic training priors of our model and the diverse chromatic and structural characteristics of real-world haze. The fixed white haze layer and reliance on homogeneous haze patterns limit its adaptability to non-white fog and layered density changes. Future research could focus on developing more flexible haze representation mechanisms and incorporating real-world data augmentation to bridge this gap, thereby enhancing the robustness of model in complex scenarios.

Conclusion

In this work, we present Alpha-DehazeNet, a novel image dehazing framework that leverages RGBA haze layer modeling to bridge physical priors and data-driven learning. By integrating transparency-aware feature extraction and depth consistency constraints, the model achieves state-of-the-art performance on synthetic benchmarks while maintaining parameter efficiency. The experimental results confirm that RGBA-based haze simulation enables precise layer separation, and the spatial-channel attention mechanisms enhance adaptability to diverse haze densities.

Nevertheless, the model faces challenges in real-world deployment. Its reliance on synthetic training data limits generalization to dense, chromatic haze, and the fixed white initialization of the haze layer introduces artifacts in non-ideal conditions. Future work could explore adaptive haze chromaticity learning, where initial RGBA values are dynamically adjusted based on real-world haze distributions, and hierarchical density modeling to better capture depth-varying fog patterns. Furthermore, integrating unpaired real-world data through domain adaptation techniques could mitigate the synthetic-to-real performance gap. To ensure a more holistic evaluation of dehazing performance, we also plan to develop a comprehensive framework with more perceptual quality indicators. These advancements would strengthen the model’s robustness while preserving its architectural simplicity, potentially improving the reliability of dehazing solutions.

Supplemental Information

Supplemental Information 1 Code.

Additional Information and Declarations

Competing Interests

The authors declare that they have no competing interests.

Author Contributions

Jin He conceived and designed the experiments, performed the experiments, analyzed the data, performed the computation work, prepared figures and/or tables, authored or reviewed drafts of the article, and approved the final draft.

Ruibin Li conceived and designed the experiments, authored or reviewed drafts of the article, experimental guidance, financial support, and approved the final draft.

Data Availability

The following information was supplied regarding data availability:

The code is available at GitHub and Zenodo:

https://github.com/GinDan/Routes.

GinDan. (2025). GinDan/Routes: Alpha-DehazeNet (v1.0). Zenodo. https://doi.org/10.5281/zenodo.15361810.

The RESIDE dataset is available at: https://sites.google.com/view/reside-dehaze-datasets/reside-standard. (DOI: 10.1109/TIP.2018.2867951).

The Dense-Haze dataset is available at: https://data.vision.ee.ethz.ch/cvl/ntire19//dense-haze/. (DOI: 10.1109/ICIP.2019.8803046).

The NH-Haze dataset is available at: https://data.vision.ee.ethz.ch/cvl/ntire20/nh-haze. (10.1109/CVPRW50498.2020.00230).

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
