# Peer review of "Alpha-DehazeNet: single image dehazing via RGBA haze modeling and adaptive learning"

_PeerJ Computer Science, doi:10.7717/peerj-cs.3036_

## Round 0.1 · original submission · Major Revisions

Please submit a revision that addresses the reviewer comments.

·

Basic reporting

The manuscript is well-written in clear, professional English, with a comprehensive literature review and sufficient field context. Its professional structure, along with well-designed figures and tables, and the sharing of raw data, are commendable. The results are relevant to the hypotheses, validating the model's effectiveness. However, for further improvement, the authors should focus on formalizing the results. Clear definitions of all terms and theorems, as well as more detailed proofs for the proposed algorithms and models, are needed. This will enhance the theoretical rigor of the study, making it more robust and facilitating future research in this area.

Experimental design

The research presented in this manuscript is original and falls well within the aims and scope of the journal. The research question is clearly defined, highly relevant, and meaningful, with a clear explanation of how it fills an identified knowledge gap in the field. The investigation is carried out with great rigor, meeting high technical and ethical standards. Moreover, the methods are described in sufficient detail, providing all the necessary information for others to replicate the study. This level of clarity and thoroughness not only validates the research but also paves the way for further exploration and advancement in the area of image dehazing.

Validity of the findings

-

Additional comments

There is a lack of some references, such as datasets and methods for 4K dehazing.

Reviewer 2 ·

Basic reporting

The related work should be enriched to show the development of image dehazing.

Experimental design

The experiments can be enriched to show the superiority of the proposed method.

Validity of the findings

The main motivations and contributions should be highlighted.

Additional comments

This paper proposes Alpha-DehazeNet, a model leveraging RGBA haze layer effect maps, which constructs a grayscale transparency map in the RGBA color space as the initial haze layer.
The weaknesses of this paper are listed as follows:
1. The introduction section should summarize the main contributions of the proposed method in a bullet-point manner. The authors should highlight the main differences and advantages of the proposed method compared to previous dehazing techniques.
2. In the related work section, the authors fail to review daytime and nighttime dehazing methods. Moreover, this paper can show the dehazing performance on nighttime hazy images.
[1] Single Image Dehazing via Multi-scale Convolutional Neural Networks with Holistic Edges
[2] VNDHR: Variational Single Nighttime Image Dehazing for Enhancing Visibility in Intelligent Transportation Systems via Hybrid Regularization (https://ieeexplore.ieee.org/document/10946240)
[3] Multi-Purpose Oriented Single Nighttime Image Haze Removal Based on Unified Variational Retinex Model Gaussian Total Variation
[4] "Nighthazeformer: Single nighttime haze removal using prior query transformer"
3. How to demonstrate the generalization ability of the proposed method? Could the authors provide the comparisons on real dataset?
4. In Table 1, could the authors add more recent state-of-the-art methods (2024) for comparisons?
5. Could the authors show the limitations of the proposed method and analyze the corresponding reasons?
6. In Figure 8, which column is your dehazed results? Could you show more visual comparisons with other methods?
I recommend revisions for this paper. This paper can be accepted after carefully revisions.

Reviewer 3 ·

Basic reporting

-

Experimental design

-

Validity of the findings

-

Additional comments

The manuscript proposes an image dehazing model, Alpha-DehazeNet, based on RGBA color space modeling, which achieves image dehazing by introducing depth consistency loss and adversarial training. However, the manuscript still suffers from the following issues:

1. The proposed method is essentially a combination of channel attention, spatial attention, adversarial training, and residual structures, resulting in a lack of significant innovation in the model.
2. The manuscript assumes the haze layer to be a white transparent image, which does not hold in real-world scenarios. This assumption may lead to artifacts when processing non-white haze. Moreover, the proposed method lacks adaptability to dynamic changes in haze color and density, making it ineffective in handling real-world haze conditions.
3. Figures 4 and 5 lack explanatory legends for each module and operation.
4. The depth consistency loss function overly relies on histograms, making it difficult to accurately capture depth information in images with significant depth variations. The manuscript does not conduct experiments on the NH-Haze dataset, which contains non-uniform haze with noticeable depth changes.
5. As shown in Figure 8, the proposed method fails in real-world scenarios. However, the manuscript only emphasizes its superiority on synthetic datasets in the experiments, without thoroughly discussing and explaining the limitations and reasons for failure on real-world datasets.
6. The manuscript uses traditional objective metrics such as PSNR and SSIM to evaluate dehazing performance, but these metrics cannot fully reflect the visual quality of images. There is a lack of consideration for perceptual quality metrics (LPIPS, FID, NIQE, PIQE, FADE).
7. The omission of detailed floating-point operations (FLOPS) reporting is significant. Evaluating performance without considering these aspects may lead to unfair comparisons.
8. The experimental comparisons in the paper are insufficiently comprehensive, lacking discussions and comparisons with state-of-the-art image dehazing methods (C2PNet [1], MB-Taylorformer [2], MITNet [3], DCMPNet [4]).

---

## Round 0.2 · accepted · Accept

Dear authors, we are pleased to verify that you meet the reviewer's valuable feedback to improve your research.

Thank you for considering PeerJ Computer Science and submitting your work.

Kind regards
PCoelho

Reviewer 2 ·

Basic reporting

This paper proposes Alpha-DehazeNet, a novel model leveraging RGBA haze layer effect maps, which defines a grayscale transparency map in the RGBA color space as the initial haze layer.

Experimental design

In the revised manuscript, the experimental design is appropriate.

Validity of the findings

Experiments demonstrate the effectiveness of the proposed Alpha-DehazeNet on several datasets.

Additional comments

After reviewing the response letter and the revised manuscript, my concerns have been addressed.